# Immunogenicity of SARS-CoV-2 BNT162b2 Vaccine in People with Diabetes: A Prospective Observational Study

**DOI:** 10.3390/vaccines10030382

**Published:** 2022-03-02

**Authors:** Eleni Papadokostaki, Anastasios Tentolouris, Ioanna A. Anastasiou, Mina Psichogiou, Evangelia Iliaki, Ioanna Eleftheriadou, Angelos Hatzakis, Nikolaos Tentolouris

**Affiliations:** 1Diabetes Center, First Department of Propaedeutic Internal Medicine, Medical School, National and Kapodistrian University of Athens, Laiko General Hospital, 115 27 Athens, Greece; elenipapadokostaki@gmail.com (E.P.); antentol@med.uoa.gr (A.T.); anastasiouiwanna@gmail.com (I.A.A.); joeleftheriadou@yahoo.com (I.E.); 2Department of Internal Medicine, University Hospital of Heraklion, 715 00 Heraklion, Greece; 3First Department of Internal Medicine, Medical School, National and Kapodistrian University of Athens, Laiko General Hospital, 115 27 Athens, Greece; mpsichog@med.uoa.gr; 4Microbiology Department, General Hospital of Heraklion, Venizeleio, 714 09 Heraklion, Greece; evangeliailiaki@gmail.com; 5Department of Hygiene, Epidemiology and Medical Statistics, Medical School, National and Kapodistrian University of Athens, 157 72 Athens, Greece; ahatzak@med.uoa.gr; 6Hellenic Scientific Society for the Study of AIDS, Sexually Transmitted and Emerging Diseases, 115 27 Athens, Greece

**Keywords:** humoral immune response, SARS-CoV-2, COVID-19, vaccine, antibodies

## Abstract

The mRNA-based BNT162b2 vaccine has demonstrated high efficacy against severe SARS-CoV-2. However, data regarding immune response in people with diabetes mellitus (DM) after vaccination with the BNT162b2 vaccine are limited. In this prospective observational study, we examined humoral immune response in participants with and without DM after vaccination with the BNT162b2 mRNA vaccine. A total of 174 participants (58 with and 116 without diabetes, matched for age) were included. Antibodies were measured 21 days after the first dose, 7–15 days after the second dose, and 70–75 days after the second and before the third dose of the vaccine. Antibodies were measured by an anti-SARS-CoV-2 receptor-binding domain IgG (Abs-RBD-IgG) assay by a chemiluminescent microparticle immune assay; values > 50 AU/mL are considered protective from severe disease. Almost 17% of participants with DM did not develop adequate humoral immune response to the BNT162b2 mRNA vaccine after the first dose; however, it was high and similar after the second dose in both participants with and without DM and remained so almost 2 months after the second dose of the vaccine. Geometric mean values of Abs-RBD-IgG were not significantly different between participants with and without DM during the study. At least two doses of the BNT162b2 vaccine are necessary to ensure adequate and sustainable immune response in people with DM.

## 1. Introduction

Coronavirus disease 2019 (COVID-19) caused by severe acute respiratory syndrome coronavirus 2 (SARS-CoV-2) is associated with high morbidity and mortality in people with diabetes mellitus (DM) [1,2]. The immune system may be dysregulated in DM affecting both humoral and cellular immunity and people with DM are at increased risk for lower respiratory tract, urinary tract, skin, and mucous membrane infections [3,4]. Previous studies have shown that individuals with DM had decreased immunogenicity to hepatitis B vaccine, while less consistent results have been reported for influenza, pneumococcus, and varicella zoster [4]. Previous studies showed that DM and glycemic control does not affect immune response [5] or the kinetics and durability of the neutralizing antibody response to SARS-CoV-2 following COVID-19 [6]. Recent studies reported conflicting results regarding humoral immune response after vaccination against SARS-CoV-2 in people with DM [7,8].

The mRNA-based BNT162b2 vaccine has demonstrated high efficacy against severe SARS-CoV-2 [9]. In this prospective observational study, we report the results of the humoral immune response in participants with and without DM after vaccination with the BNT162b2 mRNA vaccine.

## 2. Material and Methods

### 2.1. Study Design and Participants

The study was performed between May and September 2021. The first adults with DM who attended the vaccination center of our hospital were matched for age (±3 years) in a ratio 1:2 with people without DM and were included in the study. Matching for age was necessary because previous data reported that humoral immune response post-vaccination for SARS-CoV-2 declines with age [8,10]. Exclusion criteria were pregnancy, malignancy, systematic treatment with corticosteroids and/or immunosuppressant medications, renal replacement therapy and liver cirrhosis. Diagnosis and classification of diabetes was made according to American Diabetes Association criteria [11]. Body weight and height was measured, and body mass index (BMI) was calculated. Data for people with DM were collected from the medical records and the most recent (in the last 6 months) value of HbA1c was reported. Moreover, we collected data for serious adverse events post-vaccination. The study was approved by the Ethics committee of the Laiko General Hospital, Athens, Greece, and all participants had signed written informed consent. 

### 2.2. Data Collection and Outcomes

A venous blood sample was collected in Vacutainer tubes 21 days after the first dose and just before the second dose of the vaccine (T1), 7–15 days after the second dose of the vaccine (T2), and 70–75 days after the second and before the third dose of the vaccine (T3). The blood was centrifuged 10 min after collection at 400× *g* for 10 min at room temperature to isolate serum. Serum samples were then aliquoted and stored at −80 °C until the assays were performed. Antibodies were measured in serum by an anti-SARS-CoV-2-RBD IgG assay (Abbott SARS-CoV-2 IgG II Quant); it quantifies IgG antibodies against the receptor-binding domain (RBD) of the S1 subunit of the spike protein of SARS-CoV-2 by a chemiluminescent microparticle immune assay. The linear range is between 21.0 and 40,000 arbitrary units/mL (AU/mL) and according to the manufacturer, the clinical sensitivity was 98.81% (95% CI 93.56–99.94%) in samples collected ≥15 days after the positive PCR and the clinical specificity 99.55% (95% CI 99.15–99.76%), at a cutoff value 50 AU/mL [12]. Anti-SARS-CoV-2 RBD IgG assays have shown an excellent correlation with neutralizing antibodies [12]. The assay is based on chemiluminescent microparticle immune assay (CLIA) [12]. The correlation coefficient in weighted linear regression of WHO standard with the Abbott anti-RBD is 0.999, and transformation of Abbott anti-RBD AU/mL to WHO binding antibody units (BAU)/mL is feasible using the equation BAU/mL = 0.142 × AU/mL [12].

### 2.3. Statistical Analysis

The MedCalc Software (version 12.2.1.0, MedCalc, Ostend, Belgium) was used for the analyses. Results are reported as mean values ± standard deviation (SD), *n* (%) or as median value (interquartile range, IQR). Because the values of anti-SARS-CoV-2 RBD IgG titers (Abs-RBD-IgG) were skewed, geometric means (95% confidence intervals) were calculated and used for analyses. *p* values < 0.05 (2-tailed) were considered significant.

## 3. Results

A total of 180 people were recruited in the study. Data on follow up of 6 people, 4 without and 2 with DM, were not available because 2 withdrew informed consent and 4 missed or delayed the appointment for the second dose of vaccine. Five participants, 3 with and 2 without DM had confirmed COVID-19 disease through real-time reverse-transcriptase polymerase chain reaction at least 3 months before recruitment. 

A total of 58 with DM (14 with type 1 and 44 with type 2 DM) and 116 subjects without DM with full data available were recruited and analyzed. At T1, Abs-RBD-IgG > 50 AU/mL were detected in 82.8% participants with DM and in 91.4% without DM (*p* = 0.093). At T2 and T3, all participants (100%) without DM and 96.6% of those with DM (*p* = 0.110) had Abs-RBD-IgG > 50 AU/mL (Table 1).

Geometric mean values of Abs-RBD-IgG were not significantly different between participants with and without DM at either T1, T2 or T3 (Table 1).

An increase in Abs-RBD-IgG was found in both people with and without DM 7–15 days after the second dose of the vaccine. However, a decline in Abs-RBD-IgG was observed 2 months after the second dose of almost 73% in participants without and 76% in participants with DM (Table 1, Figure 1). 

In the total sample of participants with and without DM, no significant gender differences were found in geometric mean values Abs-RBD-IgG at either T1, T2 or T3; a significant negative correlation was found between age and Abs-RBD-IgG (*r* = −0.260, *p* = 0.001) at T1, but not (*p* > 0.05) at T2 or T3.

In the group of participants without DM, a significant negative correlation was found between Abs-RBD-IgG and age (*r* = −0.214, *p* = 0.031) at T1, but not at T2 or T3.

In the group of participants with DM, at T1, a significant negative correlation was found between Abs-RBD-IgG and age (*r* = −0.327, *p* = 0.020), while no significant (*p* > 0.05) correlation was found with HbA1c or diabetes duration; at T2 and T3, no significant correlation (*p* > 0.05) was found between Abs-RBD-IgG and either age, diabetes duration or HbA1c. 

No significant correlations were found between Abs-RBD-IgG and BMI in the total sample and in the participants with and without DM at either T1, T2 or T3. 

The mean age and gender of the 6 participants who were not included in the analysis did not differ significantly from the participants in the study: mean age 51.8 ± 9.4 years; 4 males and 2 females (both *p* > 0.05). A total of 75% and 88% of the participants did not have adverse events after the first dose and the second dose of the vaccine, respectively; mild adverse events were reported by the other participants like pain at injection site, fatigue, and headache after either the first or the second doses. No serious adverse events were reported.

## 4. Discussion

In this study we found almost 17% of participants with DM and 8% of those without DM do not develop adequate humoral immune response to the BNT162b2 mRNA vaccine after the first dose; however, humoral immune response was high and similar after the second dose in the participants with and without DM and remained so almost 2 months after the second dose of the vaccine. 

A pivotal phase 3 study of BNT162b2 showed similar vaccine efficacy across subgroups defined by age, sex, race, ethnicity, baseline BMI, and the presence of coexisting conditions; however, that study did not provide information for people with DM [9]. Herein we found that Abs-RBD-IgG increased by at least 15 times 7–15 days after the second dose in comparison with those just before the second dose of the vaccine in people with and without DM. These results agree with the findings of a recent study that reported a marked boosting effect on the titer of anti-S protein RBD antibody 1 week after the second dose in people with and without DM [13]. Furthermore, in agreement with recent reports that examined humoral immune response prospectively up to 6 months post-vaccination [14,15], we found a significant decline in Abs-RBD-IgG two months after the second dose of the vaccine. 

In this study, we showed that the humoral immune response to the BNT162b2 mRNA vaccine was similar between people with DM and people without DM in three different measurements (T1, T2, T3). In a previous study, a total of 86 healthy controls and 161 people with DM were enrolled, and antibody levels were measured 7 to 14 days after the first vaccination, as well as 14 to 21 days after the second vaccination (86% received the BioNTech/Pfizer, 8.7% the Moderna and 5.3% the AstraZeneca vaccine) [8]. In the unadjusted analysis the study showed highest antibody levels after second vaccination in people with type 1 DM (*p* = 0.022 vs. healthy controls and *p* = 0.013 vs. people with type 2 DM). However, after adjustment for age and sex, antibody levels after second vaccination were similar in people with DM in comparison with people without DM [8]. 

These results are in contrast with an observational study (CAVEAT study) that reported a lower antibody response to COVID-19 vaccination in people with type 2 DM having an HbA1c above 53 mmol/mol (7.0%) compared with normoglycemic individuals [16]. Nevertheless, people with DM in our study had a mean HbA1c of 51 mmol/mol (6.8%), hence, the differences might be attributed to the different glycemic status of the participants. Similarly, another study examined the levels of anti-SARS-CoV-2 IgG and neutralizing antibodies in people with type 2 DM (*n* = 81) and/or other metabolic risk factors (hypertension and obesity) in comparison with those without DM or co-morbidities (*n* = 181) [7]. Antibodies were measured to subjects a minimum of 3 weeks after the second dose of BNT162b2 mRNA vaccine. That study showed that substantial quantities of neutralizing antibodies were elicited post-vaccination, about five times the seropositivity threshold. However, in contrast to our findings and to the findings by Sourij et al., both SARS-CoV-2 IgG and neutralizing antibodies titers were lower in people with T2DM than people without DM [7,8].

The main strength of the study is that the Anti-SARS-CoV-2 RBD IgG were measured, that have shown an excellent correlation with neutralizing antibodies [12]. In addition, it is known that age is associated with lower titers of neutralizing antibodies [8,10] and herein patients with DM were matched for age with people without DM.

The study limitations include the small sample size, the single measurement of antibodies, sole use of the BNT162b2 vaccine, and assessment of the humoral component and not the T- cell immune response. In addition, we included only 14 participants with type 1 DM and the results cannot be extrapolated to all people with DM. Furthermore, the sample size is small to compare the humoral immune response in type 1 and type 2 DM. Moreover, although there were gender and BMI differences between the study groups, we do not think that this impact the results, because humoral immune response to vaccination was not associated with either gender or BMI. 

## 5. Conclusions

This study indicates that 17% of people with DM do not develop adequate humoral immune response after one dose of the BNT162b2 vaccine and a second dose is necessary to ensure adequate and sustainable immune response. After the second dose a similar response was found in both people with and without DM. Larger prospective studies are required to confirm these results and to examine Abs-RBD-IgG over time. 

## Figures and Tables

**Figure 1 vaccines-10-00382-f001:**
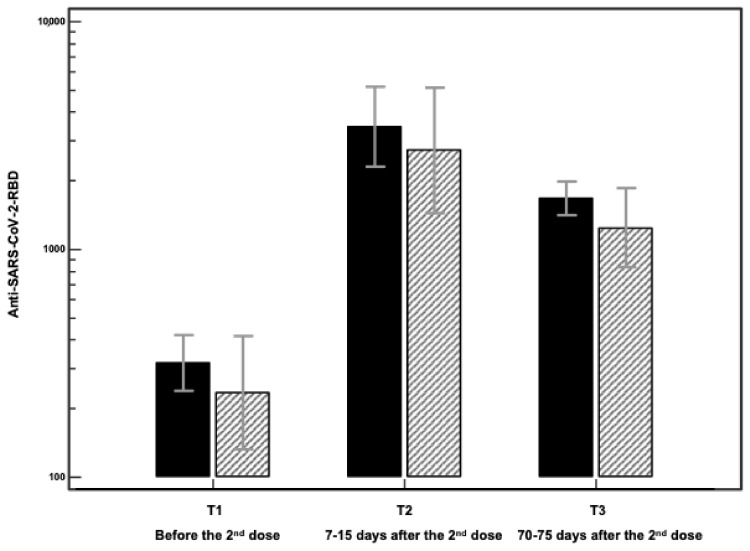
Antibody titers against the severe acute respiratory syndrome coronavirus 2 (SARS-CoV-2)-receptor-binding domain (RBD) IgG titers expressed in arbitrary units (AU), measured 21 days after the first dose and before the second dose (T1), 7–15 days after the second dose (T2) and 70–75 days after the second and before the third dose of the vaccine (T3). Black bars: participants without diabetes; hatched bars: participants with diabetes. Data are shown as geometric means (95% confidence intervals). Anti-SARS-CoV-2-RBD: anti-SARS-CoV-2 receptor-binding domain IgG.

**Table 1 vaccines-10-00382-t001:** Characteristics and humoral immune response of the study participants.

	People without DM *n* = 116	People with DM *n* = 58	*p* Value
Age (years)	51.3 ± 7.9	52.6 ± 10.6	0.430 *
Male/female, *n* (%)	38 (32.8)/78 (67.2)	29 (50.0)/29 (50.0)	0.028 **
Type of DM (Type 1 DM/Type 2 DM), *n* (%)	-	14 (24.1)/44 (75.9)	-
Duration of diabetes (years) median value (25, 75 percentile)	-	9.0 (2.0, 17.5)	-
Body Mass Index (kg/m^2^)	27.3 ± 4.9	30.5 ± 4.7	<0.001 *
HbA1c (%)	-	6.8 ± 2.6	-
HbA1c (mmol/mol)	-	51 ± 5	-
Anti-SARS-CoV-2 RBD IgG titers > 50 AU/mL at T1, *n* (%)	106/116 (91.4)	48/58 (82.8%)	0.093 **
Anti-SARS-CoV-2 RBD IgG titers in AU/mL at T1 Geometric mean (95% confidence intervals)	354.62 (268.34, 468.65)	220.10 (122.59, 395.17)	0.144 *
Anti-SARS-CoV-2 RBD IgG titers > 50 AU/mL at T2, *n* (%)	116/116 (100)	56/58 (96.6)	0.110 **
Anti-SARS-CoV-2 RBD IgG titers in AU/mL at T2 Geometric mean (95% confidence intervals)	6281.32 (5244.47, 7523.16)	5300.64 (3868.71, 7262.56)	0.350 *
Anti-SARS-CoV-2 RBD IgG titers > 50 AU/mL at T3, *n* (%)	116/116 (100)	56/58 (96.6)	0.110 **
Anti-SARS-CoV-2 RBD IgG titers in AU/mL at T3 Geometric mean (95% confidence intervals)	1677.94 (1412.94, 1991.53)	1246.77 (853.76, 1859.89)	0.173

* *p* values for comparisons between groups by independent samples *t*-test. ** *p* value for comparisons between groups by either the Chi-squared test or the Fisher’s exact test. Data are presented as means ± SD (standard deviation), *n* (%), median value (25, 75 percentile) or geometric means (95% confidence intervals). DM: diabetes mellitus, HbA1c: glycated hemoglobin, T1: 21 days after the first dose and just before the second dose of the vaccine, T2: 7–15 days after the second dose of the vaccine, T3: 70–75 days after the second and before the third dose of the vaccine, SARS-CoV-2: severe acute respiratory syndrome coronavirus 2, RBD: receptor-binding domain. AU: arbitrary units.

## Data Availability

Data are available upon request.

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
