# Peer review of "Immunogenicity of SARS-CoV-2 BNT162b2 Vaccine in People with Diabetes: A Prospective Observational Study"

_vaccines, 2022, doi:10.3390/vaccines10030382_

Round 1

Reviewer 1 Report

The humoral immune response in people with and without DM after immunization with the BNT162b2 mRNA vaccine was investigated in this brief report paper by Papadokostaki et al. In this prospective observational study, Papadokostaki et al.  examined humoral immune response in participants with and without DM after vaccination with the BNT162b2 mRNA vaccine. The authors provided a brief-documented overview of  mRNA vaccine implications in DM/without DM  and, specifically related to patient characterstics. They discovered that at least two doses of the BNT162b2 vaccination are required in persons with DM to ensure an adequate and long-lasting immune response. In addition to this, the article provides an interesting point of view regarding  its limitaion and suggest further large cohort studies.  

Moreover, minor spell and English form check is required.

Below, some comments about,

  1. What makes the sudy differe from others study worldwide like (https://doi.org/10.3389/fimmu.2021.752233
  2. Line 36 : the introduction section should properly restructured in such a way that by introducing the problem what has been done so far on COVID 19 and its intervention with and without DM cases.  Please refer ref 3, 4 and 5 line 39-45.
  3. Line 52: Authors should indicate the type of samples used in the method section under study design and participants.
  4. Line 110: Figure 1. Please indicate figure legened and label graph accordingly (y-axis??? and x-axis ) and have it as style as clearly to the readers
  5. Line 142-143: Authors stated that no other studies examined humoral immune response to the BNT162b2 mRNA vaccine in people with DM.  However, there are studies like (https://doi.org/10.3389/fimmu.2021.752233 and others ). I already suggested adding some more recent references.
  6. It would be so important if authors work includes/describes humoral immune response to the BNT162b2 mRNA vaccine compared to T1DM vs T2DM in the result section
  7. Line 157: The conclusion must focus up on the best result found in the study.As data has limited by its small number of sample size authors conculusion should be revised.

Author Response

Thank you for giving us the opportunity to revise our paper entitled “Immunogenicity of SARS-CoV- 2 BNT162b2 vaccine in people with diabetes: a prospective observational study”, Manuscript Ref. No.:  #vaccines-1594786. We also thank the reviewers for their comments and the constructive criticism regarding this work. We hope we have successfully addressed all comments as follows: 

Reviewer #1
The humoral immune response in people with and without DM after immunization with the BNT162b2 mRNA vaccine was investigated in this brief report paper by Papadokostaki et al. In this prospective observational study, Papadokostaki et al.  examined humoral immune response in participants with and without DM after vaccination with the BNT162b2 mRNA vaccine. The authors provided a brief-documented overview of mRNA vaccine implications in DM/without DM and, specifically related to patient characterstics. They discovered that at least two doses of the BNT162b2 vaccination are required in persons with DM to ensure an adequate and long-lasting immune response. In addition to this, the article provides an interesting point of view regarding its limitation and suggest further large cohort studies.  
Moreover, minor spell and English form check is required.
Below, some comments about,

Comment #1
What makes the study different from others study worldwide like (https://doi.org/10.3389/fimmu.2021.752233)
Our response:
Thank you for your comment. A paragraph with the strengths of the study has been added in the manuscript: “The main strength of the study is that the Anti-SARS-CoV-2 RBD IgG were measured, that have shown an excellent correlation with neutralizing antibodies [12]. In ad-dition, it is known that age is associated with lower titers of neutralizing antibodies [8,10] and herein, patients with DM were matched for age with people without DM.” (lines: 190-193).

Comment #2
Line 36: the introduction section should properly restructured in such a way that by introducing the problem what has been done so far on COVID 19 and its intervention with and without DM cases.  Please refer ref 3, 4 and 5 line 39-45.
Our response:
Thank you for your comment. The introduction section has been revised to: “Coronavirus disease 2019 (COVID-19) caused by severe acute respiratory syndrome coronavirus 2 (SARS-CoV-2) is associated with high morbidity and mortality in people with diabetes mellitus (DM) [1,2]. The immune system may be dysregulated in DM affecting both humoral and cellular immunity and people with DM are at increased risk for lower respiratory tract, urinary tract, skin, and mucous membrane infections [3,4]. Previous studies have shown that individuals with DM had decreased immunogenicity to hepatitis B vaccine, while less consistent results have been reported for influenza, pneumococcus, and varicella zoster [4]. Previous studies showed that DM and glycemic control does not affect immune response [5] or the kinetics and durability of the neutralizing antibody response to SARS-CoV-2 following COVID-19 [6]. Recent studies re-ported conflicting results regarding humoral immune response after vaccination against SARS-CoV-2 in people with DM [7,8].
The mRNA-based BNT162b2 vaccine has demonstrated high efficacy against severe SARS-CoV-2 [9]. In this prospective observational study, we report the results of the humoral immune response in participants with and without DM after vaccination with the BNT162b2 mRNA vaccine”.

Comment #3
Line 52: Authors should indicate the type of samples used in the method section under study design and participants.
Our response:
Thank you for your comment. The sample of the study consisted of adults with DM who attended the vaccination center of our hospital and were matched for age (±3 years) in a ratio 1:2 with people without DM (lines: 55-57).

Comment #4
Line 110: Figure 1. Please indicate figure legend and label graph accordingly (y-axis??? and x-axis ) and have it as style as clearly to the readers
Our response:
Thank you for your suggestion. The figure has been revised.

Comment #5
Line 142-143: Authors stated that no other studies examined humoral immune response to the BNT162b2 mRNA vaccine in people with DM.  However, there are studies like (https://doi.org/10.3389/fimmu.2021.752233 and others). I already suggested adding some more recent references.
Our response
Thank you for your comment. At the time this work was submitted there were no previous reports on the topic. However, now there are more studies published and this sentence has been erased. A new paragraph gas been added (lines 155-189): “A pivotal phase 3 study of BNT162b2 showed similar vaccine efficacy across sub-groups defined by age, sex, race, ethnicity, baseline BMI, and the presence of coexisting conditions; however, that study did not provide information for people with DM [9]. Herein, we found that Abs-RBD-IgG increased by at least 15 times 7-15 days after the second dose in comparison with those just before the second dose of the vaccine in people with and without DM. These results agree with the findings of a recent study that reported a marked boosting effect on the titer of anti-S protein RBD antibody 1 week after the second dose in people with and without DM [13]. Furthermore, in agreement with recent reports that examined humoral immune response prospectively up to 6 months post-vaccination [14,15], we found a significant decline in Abs-RBD-IgG two months after the second dose of the vaccine. 
In this study, we showed that the humoral immune response to the BNT162b2 mRNA vaccine was similar between people with DM and people without DM in three different measurements (T1, T2, T3). In a previous study, a total of 86 healthy controls and 161 people with DM were enrolled, and antibody levels were measured 7 to 14 days after the first vaccination, as well as 14 to 21 days after the second vaccination (86% received the BioNTech/Pfizer, 8.7% the Moderna and 5.3% the AstraZeneca vaccine) [8]. In the un-adjusted analysis the study showed highest antibody levels after second vaccination in people with type 1 DM (p=0.022 vs. healthy controls and p=0.013 vs. people with type 2 DM). However, after adjustment for age and sex, antibody levels after second vaccination were similar in people with DM in comparison with people without DM [8]. 
These results are in contrast with an observational study (CAVEAT study) that re-ported a lower antibody response to COVID-19 vaccination in people with type 2 DM having an HbA1c above 53 mmol/mol (7.0%) compared with normoglycemic individuals [16]. Nevertheless, people with DM in our study had a mean HbA1c of 51 mmol/mol (6.8%), hence, the differences might be attributed to the different glycemic status of the participants. Similarly, another study examined the levels of anti-SARS-CoV-2 IgG and neutralizing antibodies in people with type 2 DM (n=81) and/or other metabolic risk factors (hypertension and obesity) in comparison with those without DM or co-morbidities (n=181) [7]. Antibodies were measured to subjects a minimum of 3 weeks after the second dose of BNT162b2 mRNA vaccine. That study showed that substantial quantities of neutralizing antibodies were elicited post-vaccination, about five times the seropositivity threshold. However, in contrast to our findings and to the findings by Sourij et al., both SARS-CoV-2 IgG and neutralizing antibodies titers were lower in people with T2DM than people without DM [7,8]”.

Comment #6
It would be so important if authors work includes/describes humoral immune response to the BNT162b2 mRNA vaccine compared to T1DM vs T2DM in the result section
Our response
Thank you for your suggestion. The sample size is small to compare the humoral immune response in type 1 and type 2 DM. This has been added in the limitations section of the manuscript (lines 197-198): “Furthermore, the sample size is small to compare the humoral immune response in type 1 and type 2 DM”.

Comment #7
Line 157: The conclusion must focus up on the best result found in the study. As data has limited by its small number of sample size authors conclusion should be revised.

Our response
Thank you for your comment. The limitations of the study are presented in lines 194-198. The conclusion of the study has been revised (lines 203-207): “This study indicates that 17% of people with DM do not develop adequate humoral immune response after one dose of the BNT162b2 vaccine and a second dose is necessary to ensure adequate and sustainable immune response. After the second dose a similar response was found in both people with and without DM. Larger prospective studies are required to confirm these results and to examine Abs-RBD-IgG over time”.

On behalf of the co-authors,
Prof Nikolaos Tentolouris

Reviewer 2 Report

The manuscript by Papadokostaki et al. presents an interesting prospective observational study concerning the immune response of people with diabetes mellitus (DM) following the mRNA vaccine BNT162b2.

Although with some limitations, the most relevant being the focus on a single vaccine, I consider this study sound and well presented. The results are of interest, since they highlight that DM does not compromise the immune response elicited by the 2nd vaccine dose, albeit a significant effect on the 1st dose is observed.

The communication/report short format of the manuscript is tailored to the scientific question it tries to address. I see no major issues to be considered. By following, I report some suggestions:

1) In figure 1, I would add a label to Y axis (e.g. Antibody titer (A.U.) ) and I would change the X-axis label to include the assay timeline (e.g. T1 (+21 days from 1st dose) ).

2) In the Discussion section, it would be interesting to report  and compare the actual figures of RBD IgG for non-DM people enrolled in this work with values found elsewhere (ref. 10 and 11), instead of generically stating "at least 15 times..."

3) The same point holds for the observed decrease of Ab titer after 2nd dose: could the authors compare the median value they found at 70-75 days after the 2nd dose with figures reported in literature of the same sort? I understand it might be difficult to find data at exactly the same timing, but even within a few weeks that would substantiate the present work.

4) Line 106: "with AND without"

Author Response

Thank you for giving us the opportunity to revise our paper entitled “Immunogenicity of SARS-CoV- 2 BNT162b2 vaccine in people with diabetes: a prospective observational study”, Manuscript Ref. No.:  #vaccines-1594786. We also thank the reviewers for their comments and the constructive criticism regarding this work. We hope we have successfully addressed all comments as follows: 

Reviewer #2
The manuscript by Papadokostaki et al. presents an interesting prospective observational study concerning the immune response of people with diabetes mellitus (DM) following the mRNA vaccine BNT162b2.
Although with some limitations, the most relevant being the focus on a single vaccine, I consider this study sound and well presented. The results are of interest, since they highlight that DM does not compromise the immune response elicited by the 2nd vaccine dose, albeit a significant effect on the 1st dose is observed.
The communication/report short format of the manuscript is tailored to the scientific question it tries to address. I see no major issues to be considered. By following, I report some suggestions:

Comment #1 
In figure 1, I would add a label to Y axis (e.g. Antibody titer (A.U.) and I would change the X-axis label to include the assay timeline (e.g. T1 (+21 days from 1st dose) ).
Our response
Thank you for your suggestion. The figure has been revised.

Comment #2 
In the Discussion section, it would be interesting to report and compare the actual figures of RBD IgG for non-DM people enrolled in this work with values found elsewhere (ref. 10 and 11), instead of generically stating "at least 15 times..."
Our response
Thank you for your comment. The Discussion section has been revised (lines 155-190) to: “A pivotal phase 3 study of BNT162b2 showed similar vaccine efficacy across sub-groups defined by age, sex, race, ethnicity, baseline BMI, and the presence of coexisting conditions; however, that study did not provide information for people with DM [9]. Herein, we found that Abs-RBD-IgG increased by at least 15 times 7-15 days after the second dose in comparison with those just before the second dose of the vaccine in people with and without DM. These results agree with the findings of a recent study that reported a marked boosting effect on the titer of anti-S protein RBD antibody 1 week after the second dose in people with and without DM [13]. Furthermore, in agreement with recent reports that examined humoral immune response prospectively up to 6 months post-vaccination [14,15], we found a significant decline in Abs-RBD-IgG two months after the second dose of the vaccine. 
In this study, we showed that the humoral immune response to the BNT162b2 mRNA vaccine was similar between people with DM and people without DM in three different measurements (T1, T2, T3). In a previous study, a total of 86 healthy controls and 161 people with DM were enrolled, and antibody levels were measured 7 to 14 days after the first vaccination, as well as 14 to 21 days after the second vaccination (86% received the BioNTech/Pfizer, 8.7% the Moderna and 5.3% the AstraZeneca vaccine) [8]. In the un-adjusted analysis the study showed highest antibody levels after second vaccination in people with type 1 DM (p=0.022 vs. healthy controls and p=0.013 vs. people with type 2 DM). However, after adjustment for age and sex, antibody levels after second vaccination were similar in people with DM in comparison with people without DM [8]. 
These results are in contrast with an observational study (CAVEAT study) that re-ported a lower antibody response to COVID-19 vaccination in people with type 2 DM having an HbA1c above 53 mmol/mol (7.0%) compared with normoglycemic individuals [16]. Nevertheless, people with DM in our study had a mean HbA1c of 51 mmol/mol (6.8%), hence, the differences might be attributed to the different glycemic status of the participants. Similarly, another study examined the levels of anti-SARS-CoV-2 IgG and neutralizing antibodies in people with type 2 DM (n=81) and/or other metabolic risk factors (hypertension and obesity) in comparison with those without DM or co-morbidities (n=181) [7]. Antibodies were measured to subjects a minimum of 3 weeks after the second dose of BNT162b2 mRNA vaccine. That study showed that substantial quantities of neutralizing antibodies were elicited post-vaccination, about five times the seropositivity threshold. However, in contrast to our findings and to the findings by Sourij et al., both SARS-CoV-2 IgG and neutralizing antibodies titers were lower in people with T2DM than people without DM [7,8]”.

Comment #3
3) The same point holds for the observed decrease of Ab titer after 2nd dose: could the authors compare the median value they found at 70-75 days after the 2nd dose with figures reported in literature of the same sort? I understand it might be difficult to find data at exactly the same timing, but even within a few weeks that would substantiate the present work.
Our response
Thank you for your comment. The Ab titers in different studies is difficult to be compared since each method has different units of measurements. Hence, in the methods section we provide information on how to transform units used in our study to WHO binding antibody units (lines: 82-86): “The assay is based on chemiluminescent microparticle immune assay (CLIA) [12]. The correlation coefficient in weighted linear regression of WHO standard with the Abbott anti-RBD is 0.999, and transformation of Abbott anti-RBD AU/mL to WHO binding an-tibody units (BAU)/mL is feasible using the equation BAU/mL = 0.142 × AU/mL [12]”.

Comment #4
Line 106: "with AND without"
Our response
Thank you for your comment. The manuscript has been revised as follows: “An increase in Abs-RBD-IgG was found in both people with and without DM 7-15 days after the second dose of the vaccine”.

On behalf of the co-authors,
Prof Nikolaos Tentolouris